# Why Sustainable Development Requires Societal Innovation and Cannot Be Achieved without This

**Henk Diepenmaat [1,2], René Kemp [1,3,*] and Myrthe Velter [1,4]**

[1] Maastricht Sustainability Institute, Maastricht University, P.O. Box 616, 6200 MD Maastricht, The Netherlands; henk.diepenmaat@maastrichtuniversity.nl (H.D.); myrthe.velter@maastrichtuniversity.nl (M.V.)
[2] Actors Procesmanagement, Prinses Irenelaan 14-B, 3708 EL Zeist, The Netherlands; henk.diepenmaat@actors.nl
[3] UNU-MERIT, P.O. Box 616, 6200 MD Maastricht, The Netherlands
[4] Fontys Centre of Expertise on Circular Transitions, Fontys University of Applied Sciences, De Lismortel 25, 5612 AR Eindhoven, The Netherlands; m.velter@fontys.nl
[*] Correspondence: r.kemp@maastrichtuniversity.nl; Tel.: +31-4338-844-3285

**Abstract:** This paper discusses the need for societal innovation as a systemic form of innovation for sustainable development. Sustainable development requires collective action from stakeholders in the form of system building activities, which in its turn requires societal innovation. Through societal innovation, based on multiple value creation, external costs are being prevented or reduced because of innovation-oriented explorations within a wider frame (a societal improvement perspective), ascertained by the actors. This requires design thinking and proper distribution of the costs and benefits, accepted by the participants. With this paper, we hope to advance the research agenda on societal innovation based on multi-actor improvement processes and associated intentional logics, as topics that are weakly theorized in the business literature on sustainable development and the sustainability transition literature. We are critical of triple helix models and models emphasizing shared value creation because these underestimate the importance of disinterest and conflicts of interests to be managed via multiple value creation on the basis of recursive multi-actor intentionality.

**Keywords:** innovation; societal innovation; sustainability; transitions; recursive perspectivism; boundary work

## 1. Introduction

Innovation processes and outcomes are studied in different traditions. Business and market aspects are studied in global value chain studies and the business model literature [1,2], path dependence (mechanisms) in the economics of innovation [3,4], disruptive innovation in market studies [5], intra and inter-organizational aspects in organizational studies and sociology of innovation [6], innovation journeys in the specialized literature on this [7,8], and sustainability transition studies about system innovations [9–12].

In this paper, we discuss the need for societal innovation as a systemic form of innovation requiring new business models, partnerships for multiple value creation and social change. This will be done with the help of a discussion of business strategies for sustainability, the literature on sustainability transitions, and recursive perspectivism [13]. Recursive perspectivism builds on the assumption that societal practices and societal value networks ultimately derive their strength and structure from the recursive meshing of actor-specific improvement perspectives and the intentional logics following from these. Recursion allows for changing scope and level of detail of these practices and networks.

Our discussion combines and supersedes common definitions of innovation (product, process, radical, incremental, modular and architectural) and puts the discussion on innovation into a framework of value orientations and actor-based improvement perspectives. This reveals a new type of innovation: societal innovations that serve multiple needs and functions in a novel way thanks to radical, incremental, modular and architectural innovations by combining the improvement perspectives of different actors including those of citizens, business circles, consumers, governments and intermediaries. Focal points are actor-specific interpretations of their environments (their contexts) and the compound intentional logics that may follow from a mutual alignment of competencies and agreements on the sharing of benefits and costs. Institutional change is part of it because without this, actors will remain locked into traditional sectorial ways of thinking. Examples of societal innovation are circular agriculture in which the agricultural biomass is being kept into the agro-food system (as fertilizer and animal feed) and the use of wood as a carbon-negative construction material in buildings designed for long-term use, and further use of materials at the end of the buildings' lifetime. If farmers also act as stewards of biodiversity and animal well-being, even more value is created.

Societal innovation seeks both to create new value and to avoid negative costs to society as a multi-actor innovation challenge. Since societal innovation requires new value networks, it cannot be designed and implemented by a single company but requires coordinated actions within an instituted process of guided evolution informed by and catering to improvement perspectives in which value creation is maximized and costs to society are minimized. This requires design thinking and proper distribution of the costs and benefits, accepted (agreed) by the participants. In the case of societal innovation, people's ideas of a better world are taken up by businesses in business models and through partnerships with environmental NGOs and governments, whose improvement perspectives are the basis for value creation. For societal innovation, the triple helix model is a poor model because the public interest is only indirectly represented, resulting in a higher risk of a narrow value creation process, with negative costs to society and natural eco-systems. The triple helix model does not provide a sufficient basis for issues like societal fair pay, keeping money in the local economy and for rebalancing society which requires human-centered ways of organizing (*social* innovation) with an important role for civil society organizations and local government [14–16].

We discuss the innovation cube: a coherent arrangement of six types of innovation. These types are: process optimization through the use of technology, better work organization, better teamwork for existing activities, product-service innovation as a result of dedicated R&D, multiplication (scaling out), and value creation for society. Each type is characterized by a specific innovation culture and based on a certain improvement perspective and actor-network. The first three are dedicated to optimizing innovation approaches. The latter three are able to take care of exploring and pioneering innovation approaches, as is required in transitions.

The paper brings together the business literature on value creation, business model innovation, sustainability strategies and boundary spanning with the literature on sustainability transitions about system changes with sustainability benefits. To that literature (referenced in Sections 2 and 3), we add a recursive perspective on innovation and society, which is applied to societal innovation. The paper also briefly considers the transition from late modernity to plurimodernity (characterized by flexibility, resilience, diversity and social innovation). Thus far, the literature has not communicated much between each other because the unit of analysis is different. In business studies, the focus is on value creation by individual businesses, whereas in transition studies the focus is on dynamics of socio-technical change, based on a distinction between niche actors and regime actors. Whereas the first literature neglects broader dynamics, the second literature neglects the micro-dynamics of businesses and the importance of business models. Methodologically, we examine what the business and transition literature have to say about societal innovation and we offer a methodological discussion of our own on societal innovation, based on recursive perspectivism. There is no separate section on methodology because there is not so much to say other than that we are comparing relevant literature (on the basis of key publications) and using recursive perspectivism to make a number of critical

comments on Corporate Social Responsibility, Shared Value Creation and the SDG agenda as currently pursued, resulting in a plea for research on societal innovation.

The goals of the paper are:

(1)  To offer a historical discussion of the ways in which business has dealt with sustainability demands and why such attempts have been relatively futile.
(2)  To offer a more systematic approach to innovation types with attention to mutual relations.
(3)  To bridge the gap between the business model literature and the sustainability transition literature, which is done through the concept of societal innovation.
(4)  To offer a more contextualized, co-evolutionary understanding of innovation-based transformations, based on a recursive relationship between innovations, improvement perspectives and socio-economic transformations, including the transformation of modernity.

The structure of the paper is as follows. Section 2 describes the business perspective on sustainability transitions. Section 3 describes the way in which business issues are taken up in the literature on sustainability transitions. Section 4 discusses societal innovation as a distinctive innovation type. This is done on the basis of a methodological use of recursive perspectivism and a new innovation typology: the innovation cube. Section 5 discusses the triple helix model and the need for broader partnerships for societal innovation based on multiple value creation (fitting with calls for rebalancing society by Mintzberg and others). In this section, we discuss attempts at finding viable business models for a circular economy and describe new roles of citizens in the energy transition and regeneration of ecosystems with the help of the notion of societal innovation theorized in Section 4. Section 6 states the conclusions.

## 2. The Business Perspective on Sustainability Transitions

To understand the relevance of businesses for sustainability transitions, we discuss how a business has dealt with sustainability pressures, through types of innovation and its positioning within societal innovation. We review the literature that addresses the extent to which the business model concept can be linked to (changes in) the societal system.

### 2.1. A Historical Description of Business Strategies for Sustainability

Before the new discourse on eco-efficiency, which re-surfaced in the 1990s, corporate environmentalism focused on the adoption of environmental technology in order to meet regulations—a reactive approach based on economic and legal responsibilities [17]. Eco-efficiency thinking drew attention to the economic benefits of resource efficiency and waste minimization. Over time, businesses enhanced their focus to customer demands for greener and more ethical goods and services, by modifying the design of products to reduce their environmental and social impacts [18]. Increasingly, businesses adopted a pro-active environmental strategy, which included product stewardship, pollution prevention, and further resource efficiency through reuse and improved waste management. Eco-efficiency was helped by bans on landfilling and end-of-life regulations. While producing environmental benefits, these strategies of eco-efficiency and green products were criticized for not achieving enough, when considering economic growth and negative rebound effects [19–21].

The attention to more systemic responses let UNEP to advocate a green economy, where a green economy is an economy where growth in income and employment is "driven by investments that reduce carbon emissions and pollution, enhance energy and resource efficiency, and prevent the loss of biodiversity and ecosystem services" [22] (pp. 1–2). Social issues are taken up as part of corporate social responsibility and the concept of an inclusive green economy, aimed at improving human well-being and building social equity while reducing environmental risks and scarcities [23]. A concept receiving much attention today by policymakers and business is the circular economy concept. The circular economy is defined and taken up in different ways. In Europe, it refers to resource efficiency and opportunities for business, in China to environmental protection more generally [24–26]. A contemporary definition is presented by [24], who define a circular economy as "a regenerative system

in which resource input and waste, emission, and energy leakage are minimized by slowing, closing, and narrowing material and energy loops" [24] (p. 765). This can be achieved through long-lasting design, maintenance, repair, reuse, remanufacturing, refurbishing, and recycling. The circular economy is intended to replace the linear model of make, use and dispose, and requires novel forms of production and consumption [27]. Towards this end, the European Commission adopted an ambitious Circular Economy Package, with legislative proposals on waste and a detailed action plan with measures covering the material cycle of production, consumption and waste management and the market for secondary raw materials [28].

In the literature on a green, more circular economy, there is an increasing interest for boundary-spanning concepts such as business model innovation, multi-actor partnerships and social change [24,29,30]. Whereas social issues are usually an add-on in the literature on the greening of business, they are prominent in the literature on social enterprises and social innovation [31,32]. Boundary spanning, business model innovation, social enterprises and multiple value creation are given little attention in the literature on sustainability transition, focusing on the socio-technical transitions to renewable energy, sustainability mobility, sustainable agro-food, because the literature is more concerned with patterns of socio-technical change than with business strategies for sustainability.

## 2.2. Boundary Spanning and Business Model Innovation

Due to their boundary-spanning nature, business models are central to systemic business strategies for sustainability [33–35]. The business model can be understood as a conceptual tool to understand 'how a firm does business', by describing the rationale of how a firm proposes, creates, delivers and captures value [33,36,37]. The value proposition describes the product/service offering to a firm's stakeholders, the value creation and delivery relates to key-activities, resources and the actors in the delivery network, and captured value relates to cost- and revenue structures [19,30]. Business models are boundary spanning as they link the focal firm (through activities such as design, production, supply chains, partnerships, and distribution channels) to its network (such as suppliers, partners, customers) [38,39]. These links are established based on the underlying value creation logic, being dominated by "the benefits related to costs" [30,40,41]. Any change in the business model is a form of business model innovation, which involves a process of "changing the business model (by creating, diversifying, acquiring or transforming) in response to internal and external incentives" [30] (p. 200). The intensity of business model change affects its sustainability potential.

Sarasini and Linder [38] make a distinction between business model innovation as a subject of innovation and as a vehicle for innovation. Traditional management literature considers business model innovation for innovations focused at changes in products/components (modular) or process (architectural) [42]. Recent literature adopts a more systemic view by considering business model innovation as a vehicle for innovation, dedicated to aligning the firm-focus with inter-organizational and societal improvements in the business ecosystem [43,44]. Schaltegger et al. [45] define three strategies for embedding sustainability into business model innovation:

(1)　the defensive strategy, which focuses on reducing risks/costs to maintain business as usual
(2)　the accommodative strategy, which focuses on ameliorating the business model to reduce impacts
(3)　the proactive strategy, which focuses on completely new designs of the value logic.

Business model innovation as a defensive strategy focuses on creating firm value through modular and architectural changes. Accommodative strategies additionally focus on exploring win-win situations in its external network to reduce negative impacts [40]. Pro-active strategies, such as business models for sustainability (BMfS), correspond with the business model as a vehicle for innovation, dedicated to aligning the firm-focus with societal improvements [38,39,45]. BMfS integrate sustainability principles in the core logic of businesses and rethink the value proposition, creation, delivery and capture mechanisms in order to maximize societal and environmental benefits in addition to economic profit [33]. After reviewing the literature, BMfS are defined as "business models that incorporate

pro-active multi-stakeholder management, the creation of monetary and non-monetary value for a broad range of stakeholders and hold a long-term perspective" [46] (p. 409). Examples of BMfS mentioned by the authors are circular business models [47], social enterprises [48], bottom-of-the-pyramid solutions [49], and product-service systems such as lease or performance models [50]. BMfS thus span boundaries even further as they are rooted in novel value orientations, by proposing, creating and capturing value as a multi-relational, multi-dimensional and multi-level concept rather than economic profit maximization for a selected group of stakeholders [51,52].

### 2.3. Business Models and Societal Transitions

While academics and practitioners are increasingly studying the innovation of BMfS [30,46], its extensive boundary-spanning nature poses significant challenges for implementation and upscaling [53]. Business model innovation for sustainability (BMIfS) differs from traditional business model innovation, in the sense that it not only requires enhanced boundary changes of the focal firm as a basis to develop BMfS, but also of its external network, or the 'business ecosystem'. In the strategic management literature, the term 'business ecosystem' is used to describe the economic and social landscape of which an individual business is a part and in which it evolves together with other businesses [54,55]. Actors in the business ecosystem can be required to shift their value orientations and innovate their business models as well, possibly resulting in the adoption of novel activities, roles and revenue structures [43,51]. Transition studies on the other hand, speak of the societal context of businesses, defined by Loorbach et al. [17] (p. 3) in an actor-centered way as: "the actors involved in a certain domain, (power) relations between them and dominant practices and mindsets". As noted by sustainability transition scholars and ecological economists, BMIfS require changes in the societal context in various ways: in institutions of permissiveness (regulation) and incentives for sustainable and unsustainable action, which depend on policy decisions created in systems of governance and other sources of change. Spanning the boundaries of BMIfS toward societal scales poses significant coordination challenges and involves the creation of multi-actor partnerships and social change [38,53]. Still, the BMIfS literature typically stops after saying just that, it does not offer a theorization of the governance and steering aspects of the transformative change via new partnerships, it is silent on the role for citizen groups, knowledge intermediaries, and the creation of commons to safeguard the public interest.

The exploration of inter-organizational boundaries for business model innovation, and especially business model innovation for sustainability, comes with a number of challenges. First, it is often not in a business's DNA to extend their network with unfamiliar stakeholders and subsequently to link business results to societal results [56]. Additionally, the systems approach of sustainability leads to "a complex network of systems interlinkages, difficult trade-offs, and powerful feedback loops within the political, business, and natural environments" [57] (p. 7). To address these challenges, a recent stream of research focuses on experimentation with sustainable business models. Sustainable business model experimentation happens in niches wherein value creation strategies are identified, tested and evaluated [43,58]. Business Model Innovation can, therefore, be seen as a process which starts as a niche-level activity, but through strategic and reflexive activities in multi-actor collaboration could function as a source of socio-technical change. However, the transformative potential of business models has its limitations. While BMIfS is able to elicit required changes in the market, and to some extent in its political sphere, it alone is unable to sufficiently alter political, regulatory and market structures as such structures are resistant to change and a matter for politics [38,59,60].

While strategic management literature is expanding its scope to a systems approach, it is still based on the utilization of current organizational boundaries and action spaces and falls short in a focus on novel mindsets towards value and stakeholders.

## 3. The Sustainability Transition Perspective on Value Creation

The sustainability transition literature branched out from innovation studies, specifically evolutionary approaches to innovation and science, technology and society studies. Transitions

as socio-technical regime shifts [61,62] or system innovation [63] are based on the work of Nelson and Winter [63] and Dosi [64] about technological regimes and paradigms, to which it offered a more developed socio-technical perspective. Next to offering schemes for analyzing socio-technical dynamics, the literature includes contributions offering guidance on the steering of sustainability transitions through strategic niche management [65], transition management [66–68] drawing on complex system theory [69] and steering as socio-technical alignment and modulation of ongoing dynamics [65]. Transition studies focus on the dynamics of socio-technical change rather than the dynamics of business. It is concerned with mechanisms and patterns, and not with the strategies of individual companies. At the heart of transition research is the distinction between regime-improving innovations and regime-altering innovations and processes [9,61]. Transitions are the long-term result of landscape pressures on socio-technical regimes, niche innovations breaking out, socio-technical alignment, new circumstances and new and altered institutions [70]. They are not the result of collective choice, because the steering is an emerging phenomenon: "politics is the constant companion of sociotechnical transitions, serving alternatively (and often simultaneously) as context, arena, obstacle, enabler, arbiter, and manager of repercussions" [71] (p. 71).

In the transition literature, the value propositions of new products and the mechanisms of value delivery and appropriation (as central elements of any business model) are not theorized. Neither is the element of multiple value creation because the literature is interested in actors at the network level, resource mobilization processes, the role of visions, dynamics of expectations (including hype and disillusionment cycle) and socio-technical alignment processes (as topics which are typically backgrounded in business studies) in relation to meeting functional needs like energy services, mobility and food consumption in more sustainable ways. Compared to organizational scholars, sustainability transition studies pay more attention to non-business actors, especially users, intermediaries, protest groups and the institutions that are shaping business choices and socio-technical outcomes, but the processes through which frames of evaluation of different actors are being combined and integrated into new value propositions, lack depth and proper theorization. In the literature on sustainability transitions, the term system innovation is favored over the term societal innovation, where system innovation refers to a change from one socio-technical system to another [9]. To us this definition is overlooking that societal change can be based on a wide variety of changes, some of which are more radical and transformative than others. Societal innovation suffers less from this problem in that it can refer to a wide range of (re)configurations for meeting material and immaterial needs differently, with an important role for social innovation. An example is renewable energy use by energy cooperatives, in which the latter is viewed as something which is desired not only from the point of technology but also wanted socially by those involved (based on feelings of duty, relatedness and making a contribution to a better world together with others). Societal innovation goes beyond meeting energy demand differently by involving social change and new organizational forms, which are used not only in energy but also in other domains, for example, agriculture, through new forms of ownership of farm-land and the use of urban gardening for food production by citizen groups with those practices catering to needs for social interaction and learning skills, next to producing food and making cities more climate-resilient and biodiverse. Societal innovation transcends sectors.

In the business literature and sociotechnical transition literature, the remaking of the economy and society is under-thematized. System innovation is juxtaposed to system improvement, as an unhelpful dualism. For those reasons, we prefer the concept of societal innovation, as a concept that can be applied at different scales (with different levels of concreteness) in a less mono-functionalist way.

In an important contribution on collective system-building activities [72], transitions scholars whose work is grounded in the Technology Innovation Systems (TIS) approach, engaged themselves with the strategic management literature on entrepreneurial infrastructure [73,74] and business eco-systems for new products with societal benefits. This resulted in a framework for collective system building, which was applied in an embedded case study in the field of the Dutch smart grid sector (Figure 1).

**Figure 1.** Overview of the strategic framework for system building and its system-building activities. Adopted from [72].

The framework consists of four clusters of factors relevant to the evolution of a technology innovation system such as biogas or biorefineries. The first category Technology development and optimization comprise activities in relation to the development and further optimization of new technology, such as testing and the creation of supplementary products and services. The second category is that of Market creation. This includes business model innovation and activities around market launch such as (collaborative) marketing and obtaining support from government and other types of actors. Market creation is important for getting feedback from users, awareness-raising and creating a constituency behind a product. In the literature on innovation management, such aspects are typically considered (even foregrounded) and discussed. This is less true for category three: the Stimulation of socio-cultural changes, comprising things such as educating users, developing competencies at collaboration and communication, the fostering of the generation of a pool of skilled labor. The fourth cluster is about the meta-task of connecting the other system-building activities. This comprises activities and processes such as industry standard-setting for products, formulation of visions and common goals for a technology innovation system, and the allocation of roles and tasks amongst industry actors and other relevant actors.

The scheme is applied to the case of smart grids in the Netherlands. What the authors found is that the entrepreneurs and entrepreneurial managers indeed engage in system-building activities of different sorts, but in an ad hoc way: "They [the entrepreneurs] are aware that they have to solve problems and overcome barriers at the system level. However, they do not strategically plan system-level changes. Instead, they formulate their strategies at the firm level and collaborate in networks to achieve their companies' objectives. As a result, they intuitively engage in system-building activities which tackle problems at the system level. However, most interviewees stated that a more strategic approach to collective system building would lead to faster diffusion and adoption of their new technology. To summarize, entrepreneurs and entrepreneurial managers are aware that they have to build a system, and they consider system-level changes. Yet, in most cases, their strategic focus is on the firm level." [72] (p. 234). This highlights the need for coordination and social innovation for systemic change, a topic to which we will discuss in Section 4 on societal innovation. In TIS and most of the transition literature, social innovation is studied primarily in relation to technological innovation (not as a focal phenomenon itself). In so doing, important aspects of change (those having to do with ethics,

government responsibilities, trust, procedural and distributional justice) are backgrounded. This is less true for socio-institutional approaches, amending weaknesses of socio-technical and socio-ecological transition approaches [75–77].

The article 'The three roles of business models in societal transitions' offers a theoretical discussion of business models in societal transition dynamics [59]. Next to discussing four roles of business models, they come up with three types of business models. The first type is that of business models that are part of the socio-technical regime (as a regime adaptation). An example of this is the selling of renewables generated electricity by utilities. This is done by the big four utilities in Germany, with one rebranding itself as a renewables company. The second type is business models as an intermediate between the technological niche and the socio-technical regime. An example of this is the business model of DZ-4 (a start-up company) who leases PV systems to households. Next to paying for the installation, it also takes care of the installation and maintenance, and it takes care of the selling of excess electricity via the grid and bringing extra renewable electricity via the grid to people home when this is needed. The 15-year tariff (offered by the government) protects households against rising energy prices, as an additional benefit. The third business model discussed in the article is that of a non-technological niche innovation such as the virtual power plant: the selling of electricity generated by small products in a bundled way to utilities and big users on the spot market, via an algorithm that calculates when it is most profitable to produce electricity and when it is least expensive to consume. To smaller producers, it offers a convenient way to access the market, and to existing network operators it offers a means to guarantee greater stability in the grid. As in the two other cases, value is being created for all parties concerned.

## 4. A Recursive Perspective on Innovation and Society

In this paragraph, we offer a recursive perspective on innovation, especially societal innovation. The section is based on the work of one of the co-authors [13,78,79] about understanding complex multi-actor situations and identifying options for multi-actor process management. Different from the traditions described in Sections 2 and 3, it does not start from the conceptualizations of the two sustainability literature pieces, but from the recursiveness of improvement perspectives in relation to the intentional creation of multiple 'real-world values'. Improvement perspectives are basic representations of the different improvement potentials as experienced and eventually executed by specific actors. The recursive perspective is based on a social constructivist view of the context. Different actors are likely to experience different parts of "a context", and to interpret "shared" parts differently. Improvement perspectives are based on values for justification and valuations of identified possibilities and threats. Existing societal practices reflect (implicit) valuations of the past, while the valuation of new (future-oriented) practices is something that needs to be done explicitly anew by the actors concerned. This value is grounded in actor-specific experiences of what is considered to be good (ethics) and what is considered to be true (possible and right). Working from pragmatic premises and drawing on a societal notion of intentional logics that is based on knowledge theoretical foundations, Diepenmaat has developed a methodological framework (depicted in Figure 2 and inspired by [80]) and subsequently augmented this framework for societal innovation. The methodological framework is based on five layers.

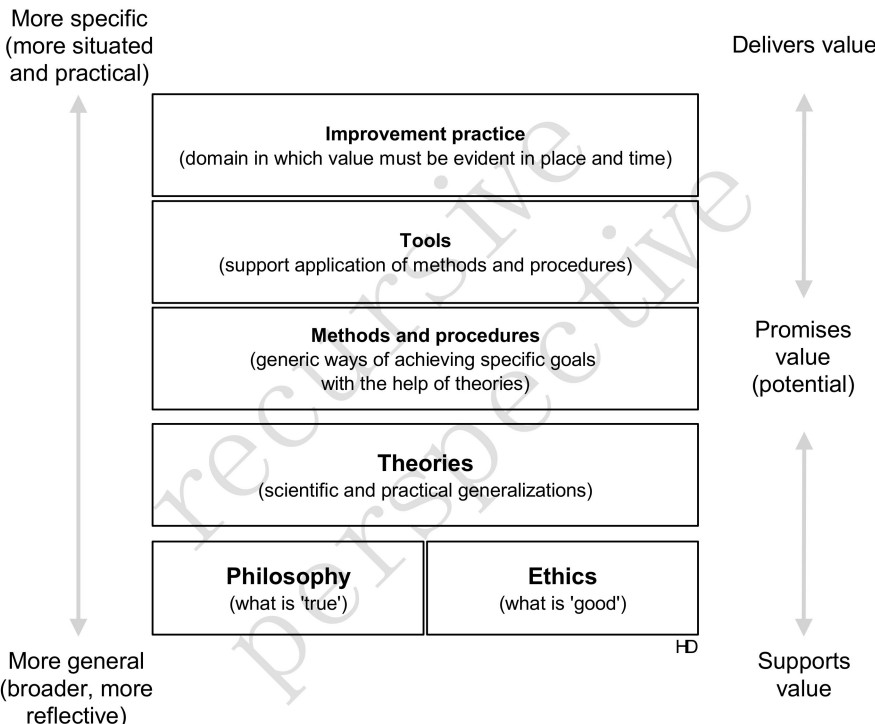

**Figure 2.** The pragmatic methodological framework of recursive perspectives. Adopted from [13,78].

At the bottom of the framework, the level of philosophy (layer 1), there are the implicit and explicit considerations of value that delimit and explain a.o. what can be observed and what cannot, what can be done and what cannot, what is considered to be an improvement and what is not, and therefore what types of value can be generated and what can not. The theory level of the general framework (layer 2) comprises of concepts and the relations linking them in conceptual wholes (networks of concepts, theories). These concepts and theories intend to support and guide the interpretation of cases at hand, and the development of methods and procedures (layer 3). The framework considers users of the methodology as reflective practitioners using these theories and methods. These methods and procedures should be supportive in achieving the intended type of goals better. Methods and procedures may be provided with tools (layer 4). Tools are of a purely operational nature: they do not change methods and procedures conceptually, they merely enhance, improve, speed up the use of methods and procedures while operating in the domain of application (layer 5) in order to generate the desired values.

The layers are coupled and require each other: their relationship is mutually supportive and symbiotic, rather than hierarchical. The layers need filling out in ways dedicated to the domain of application and in close relationship to each other.

### 4.1. Societal Innovation as a Recursive Multi-Actor Improvement Process

For *societal innovation*, the domain of application is the societal canvas, rather than a sector. Societal innovation is actor-based: it starts from the improvement perspectives of different actors and it is exactly this aspect that is fundamental to achieving intentional societal change: every actor favors change fitting with their improvement perspectives, so the innovation challenge is to create value towards those [13,78]. *As societal innovation emphasizes the meshing of improvement perspectives, sustainable development needs societal innovation and cannot be achieved without this.* Taking up sustainable development in ways that are marginally important to actors will not bring much. Perspectives constitute (radical) actors, and actors constitute societal practices, vice versa. Within this three-layered pragmatic view of society, in which perspectivist canvasses and societal practices are coupled via actors (Figure 3), collective intentional logics emerge from intentional multi-actor improvement processes.

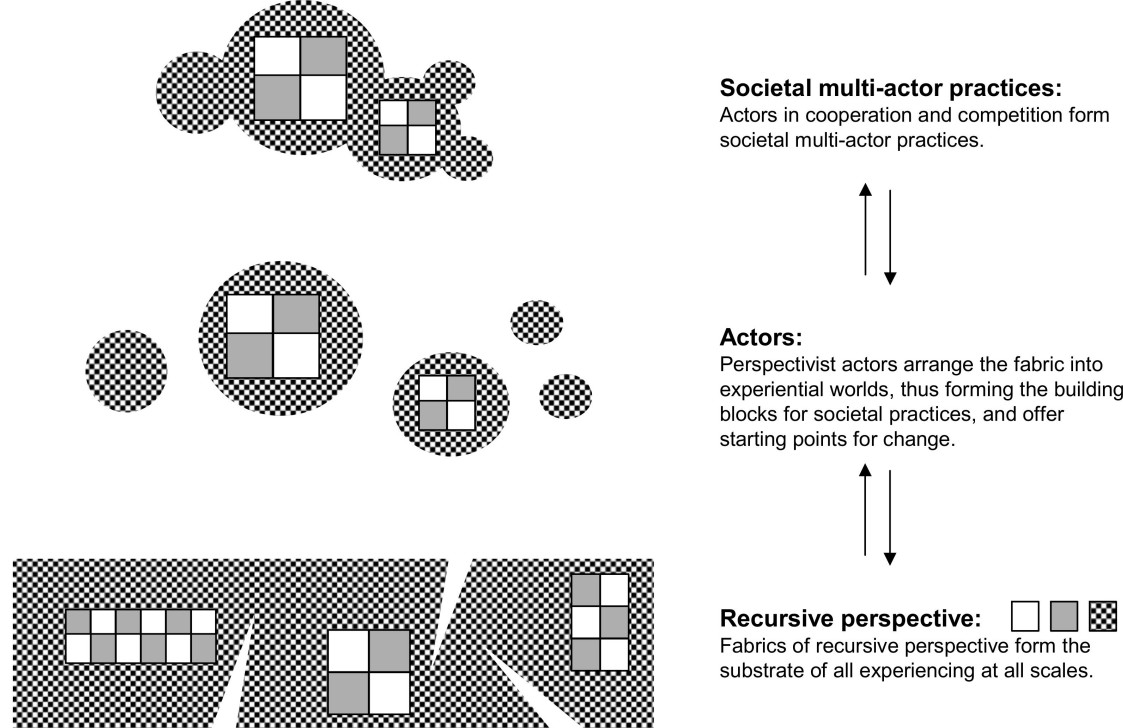

**Societal multi-actor practices:**
Actors in cooperation and competition form societal multi-actor practices.

**Actors:**
Perspectivist actors arrange the fabric into experiential worlds, thus forming the building blocks for societal practices, and offer starting points for change.

**Recursive perspective:**
Fabrics of recursive perspective form the substrate of all experiencing at all scales.

**Figure 3.** A three-layered view of society. Adopted from [13].

## 4.2. Recursive Perspectivism

Perspectives are highly *recursive* entities, each consisting of a situation before, a process and a situation after. They typically can be broken down in smaller perspectives, and several of them taken together may result in a new, larger one. The same holds for actors and societal practices. As a result, the intentional logics of different actors typically mesh in recursive ways: the final situation of a first improvement perspective may be (part of) the initial situation of another. For example, in a door renovation project, the sanding of a door is followed by the painting. Or larger improvement perspectives can be broken down in smaller ones, vice versa. For example, repairing a flat tire of a bicycle or renovating a door can be broken down in several smaller intentional steps, vice versa. This is why we talk of recursive perspectivism. Recursion has several well-defined degrees of freedom. When used in modelling networks of intentional actors (reflective intentional modelling), these degrees of freedom in combination result in a flexible means to obtain a better understanding of societal practices and their innovation. For an in-depth discussion and early examples of this, see [78], Parts V and VI.

Actors emerge in, arrange and vanish in networks of perspectives giving rise to intentional logics. Due to its flexible ontological nature (in terms of theoretical assumptions), recursive perspectivism (the recursive perspectivist view of society) has methodological benefits in improving societal practices. Both the largest and most detailed societal practices can be modelled as networks of intentional logics, possessed by actors, carrying perspectives. Whereas for example the use of a plastic straw, a landfill or a drowning archipelago are confined to specific systemic levels, the notions of improvement perspectives, actors, societal practices and intentional logics are relevant at any systemic level. They are abstract concepts that do not vanish or emerge when changing systemic level. This is intimately related to being intentional, i.e., with being human. This is the reason why we use recursive perspectivism as a pragmatic view in societal improvement: it is independent of systemic level and meaningful at al systemic levels (see also [13] (p. 32)). As a consequence, the recursive perspectivist view of society can be used in several quite different domains of societal innovation.

Societal innovation as an intentional activity requires the bringing together on societal scales (the scale of societal practices) of what is thought to be "true" and what is thought to be "good" with the help of tools, methods and theories for new doings. This bringing together, a multi-actor process in itself, requires the development of a flexible hybrid of true and good: a multi-actor improvement perspective (a specific type of perspective). Improvement perspectives are representations of potential improvement opportunities. Table 1 presents empirical examples to illustrate relevant aspects around determining what is good, to create a multi-actor improvement perspective for three cases. Since the effects are co-produced, many aspects need to be considered and managed.

**Table 1.** Empirical examples of multiple value creation based on multi-actor improvement perspectives (source: Authors).

| | |
|---|---|
| **Rondeel Eggs** | In the case of Rondeel eggs, five functionalities are being combined: animal well-being, compact use of space, the collection of eggs should be labor-extensive, efficient removal of chicken manure, affordable price for consumers. With the help of design thinking and multi-actor management, the five requirements were all met. Animal well-being had to be determined, which was done on the basis of animal behavior studies and discussions with environmental groups about animal well-being. The support of animal well-being groups helped to win over support from consumers and the higher retail price for the eggs paid for the extra costs in connection to the newly built system and use of more healthy chicken feed. The eggs are packaged in compostable package material based on potato flour, in an eye-catching design. The eggs are sold directly to a big retailer (AH) where they are part of the "pure and fair" product line, in which they are sold for an extra price of 10 cents per egg. |
| **Sustainable Packaging** | As the biggest market segment in the plastics industry, the packaging is under much pressure because of (a.o.) oceanic plastic pollution and the emerging awareness of the negative impacts of micro and nano plastics. In determining the meaning of sustainable packaging, different approaches are possible. The firm could start with rethinking individual packaging. This, however, leads to an incremental improvement on the product level. Spanning the boundary to product-packaging combinations enhances the transformative character of innovation. More sustainable individual packaging and product-packaging combinations can be achieved by cross-sectoral collaborations, in which producers and packaging experts play an important role. However, when lifting the discussion to the level of the unsustainability of our production-consumption chains and their enormous societal benefits, the true societal scope of innovating our current production-consumption chains, including packaging, comes to the surface. Turning our current production-consumption systems into sustainable systems while maintaining its societal benefits requires the additional involvement of consumers, governments, companies, knowledge institutes and intermediaries. At this societal system level, producers and packaging experts are participants in a far larger team, rather than in charge. The question is who should take up the glove to organize and further these massive cooperative societal innovations. |
| **Wood as Construction Material** | Concrete is responsible for 8% of $CO_2$ emissions, far above those from aviation. In houses and many other buildings, concrete can be replaced by wood-based (or wood-supported) constructions. Combined with forestation, wood helps to reduce carbon by capturing carbon from the air (thus serving as a negative carbon resource). When done properly, wood use could serve goals such as climate adaptation and eco-system improvement, when not done properly, it could result in unattractive forests with low resilience. Sustainable wood use is thus connected with carbon compensation reforestation schemes, sustainable forestry and deliberate attempts to bring benefits to local communities living in and around forests, something that requires special attention, care and solution design thinking [81]. In addition, houses can be designed for re-use of materials and for more communal ways of living. Different configurations are possible, allowing for place-sensitive solutions that cater to local needs and circumstances. |

### 4.3. Improvement Perspectives and Intentional Logics

Improvement perspectives make sense of the intentional logic [82] of actors in the following way: intentional societal actions require the presence of improvement perspectives, and the development of value-generating societal practices requires the networking of several improvement perspectives of actors (see [78,82] for examples of this) who differ in what they consider to be "true" and what they consider to be "good", both of which define the possible and desirable. The concept of an actor enables a grouping together of individuals on the basis of the similarity and connectedness of their improvement perspectives, or a division into smaller actors of larger actors on the basis of the differences between their improvement perspectives. From a philosophical point of view, similarity and connectedness of perspectives even are what defines an actor [78]. Actors making up complex players fields give rise to several theoretical constructs for societal innovation, of which two examples (the innovation cube and the backbone) are discussed later in this paper.

For an actor to become intentional and a perspective to become action-guiding, both an environment (which makes the possibility to execute the perspective true) and an intention (which makes the possibility to execute the perspective good) must be available. The very moment that such a bidirectional modelling relation is available, intention (top) and environment (bottom) become coupled (Figure 4). Typically, several bidirectional couplings are competing in both strength and completion, with one winning. As a result of this coupling, the possessor of the mental model will become an intentional actor and make an attempt to realize the desirable situation (upper right) by executing the available script (middle) in the environment that is present (lower left).

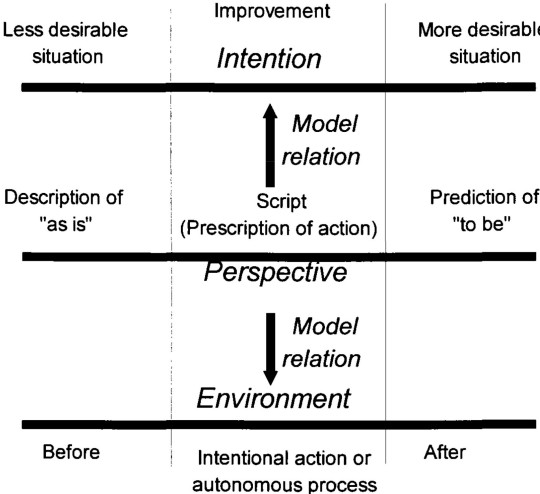

**Figure 4.** Intentional acting requires a bi-directional coupling of an improvement perspective. Adopted from [78].

Once an improvement perspective becomes bi-directionally coupled, an actor assumes that he is able to make true (in his environment) what is good (according to his intentions). At that very moment, the improvement perspective functions as an improvement potential: the actor is able to improve. Obviously, the coupling must cut ice in order to gain some value: each of the three parts may be inadequate. The result of a growing bi-coupling is the simultaneous emergence of an intentional actor who will attempt to carry out the improvement perspective. An intentional logic provides the driving force: the difference between the 'less desirable situation' and the 'more desirable situation' causes a tension, a potential, an urge, a will in the actor possessing the perspective, and the script offers him a route, a path that allows him to change the environment. The very moment that both will, and path are developed sufficiently, the actor will act: he will try to collect the benefits promised by the improvement perspective (this is the self-executing nature of bi-directionally coupled improvement perspectives).

As soon as a bystander of an action is knowledgeable of the improvement perspective of the actor (in a situated way), he will reflectively understand the intentional logic driving the action at that very

moment. This is reminiscent of the rationality principle of [83], which states that an agent will act in the most adequate way according to the logic implied by the objective situation at hand, although the intentional logic model does not presuppose an objective physical reality (Popper). It is also reminiscent of the principle of rationality of [84] that assumes pragmatically bounded knowledge grounded in such an objective reality (see also the bounded rationality principle of Simon). Our intentional logic does not need the postulation of an underlying objectively knowable reality (although in many cases such a postulation may be helpful). It merely assumes that an improvement perspective intentionally models some environment according to the actor. This offers the possibility of a reflective attitude towards societal practices without assuming some undisputed underlying truth or shared value: actors may have intentions with respect to an environment, and by acting the actor may improve. The only requirement of an intention is that it represents an improvement in an environment (in a physical or communicative or mental or any desired sense) according to the actor at stake.

Of course, the specific improvement perspectives that apply for specific actors are highly context-dependent, which means that actions are shaped by the different understandings of the situational context. Different situational contexts will lead to different compound intentional logics and multi-actor actions that stem from these. The rise of a compound intentional logic (combining different improvement perspectives) will prompt the actor to rise and act (not in a simple way, ambivalences, lack of trust and internal conflicts may act against new action). Context 'acts' via the awareness of improvement perspectives of intentional actors. It is important therefore to view the situation from the point of view of actors, which is why calls based on the SDG (as an external improvement perspective) will have little influence if they conflict with an actor's improvement perspective.

An interesting pragmatic feature of intentional logics is that constructing and changing multi-actor networks in order to realize societal innovations and more sustainable futures does not require agreement on environments, acts and intentions to a large degree. Only at the interfaces between actors (the places where radical actors share partial environments), some agreement is required. According to several sustainability scholars, shared goals and shared values are believed to be a crucial missing link in achieving sustainable outcomes (e.g., [40]). This sharing, however, is in contrast with the cultural differences and functional specializations that are so typical (both in beneficial and detrimental manners) for our advanced late modern societies.

We are not disputing the existence of shared values (everyone is in favor of environmental protection and fair pay), but we are saying that those values in themselves are not the primary shakers and movers of behavioral change and investments of firms. They interact with improvement perspectives that are more proximate to decision making (such as costs and practical matters).

Complex societies thrive on bringing differences into coherence, not on elimination of differences. Rather than attempting to further shared values, our methodological framework for societal innovation makes a point of respecting differences and bringing them into coherence as flexible components in intentional multi-actor networks. The only thing that needs to be shared at the level of societal practices is the acknowledgement that actors require each other in realizing their own needs and wishes and may help each other in this respect.

The presentation and discussion so far allow us to give a general definition of societal innovation on the basis of improvement perspectives. Intentional improvement is described as "consciously making true what is good" which requires acting according to an improvement perspective. Societal intentional improvement requires that the improvement perspective is multi-actor, typically in all its three parts. However, merely *using* an improvement perspective does not cover what we colloquially and professionally call innovation, about which we know the following that it requires determination, that results are not guaranteed and if successful creates value at multiple places (diffusion and repetition). Innovation, therefore, is *a quest for robust improvement* and successful innovation results in new or better improvement perspectives. Societal innovation, therefore, is a multi-actor quest for robust multi-actor improvement perspectives.

On the basis of recursive perspectivism, it is possible to fill out the methodological framework dedicated to societal innovation on all the five layers of Figure 2. As a whole, the framework for societal innovation supports multi-actor quests for robust multi-actor improvement perspectives. On the philosophy layer, improvement perspectives combine what is considered good and true, causing (collective) actors to emerge from the societal canvas on the very moment of potential action. On the theory layer, societal practices are to be understood as recursive multi-actor networks, driven by intentional logics on the basis of improvement perspectives. Several *theories* were developed on top of this general basis, notably the 'innovation cube' and the 'backbone'. On the *methods layer,* an intentional multi-actor modelling language and a stakeholder analysis method (the PAIR analysis) have been developed, presented in [13,78,82].

Recursive perspectivism is not culturally specific, but the societal innovations which are likely to emerge from multi-actor perspectives will differ from one culture to another (because of cultural histories, sensitivities and circumstances). Absence of trust will act as a negative factor to community-based action. Distrust of government works against partnerships with government. Systems of domination are likely to be reproduced, except when they are contested and worked around (through coordinated action based on improvement perspectives of a coalition and clever strategies).

*4.4. The Innovation Cube*

The actors making up a societal practice are distinguished on the basis of the similarities and differences of their perspectives, which are rooted in the experiential worlds of these actors. The specific way in which experimental worlds of different actors are the same or different can be visualised in three dimensions, resulting in a three-dimensional space that characterizes the perspectivist structure of the multi-actor situation at hand. The dimensions are:

- the number of different actors, i.e., actors with different experiential worlds (*p* for pluriformity, for example, bakers, political parties in a debate, and butchers are different actors),
- the mean number of actors with similar experiential worlds (*s* for similarity, the number of bakers, the number of politicians in a political party, and the number of butchers), and
- the average perspectivist scope of these different experiential worlds (*u* for unity).

A societal practice now can be positioned qualitatively in this space on the basis of those dimensions (pluriformity, similarity and unity). Societal change amounts to moving the societal practice in the resulting space. Example shifts are: the number of different actors may change (pluriformity), or the mean group size of these actors may change (similarity), or the mean perspectivist scope of these actors may change (unity). The three dimensions are discrete inversely proportional with respect to the total number A of perspectives present{XE "three dimensions of perpectivism:as actors"}: $A = \mathbf{p} \cdot \mathbf{s} \cdot \mathbf{u}$. A structure-effort correspondence hypothesis allows for drawing iso-effort planes. Figure 5 presents a two-dimensional example: in this case, the planes become lines, resembling the altitude lines on hiking maps.

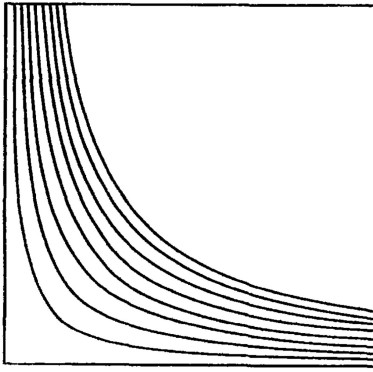

**Figure 5.** The discrete inversely proportional relations between a number of different actors and the mean number of perspectives they possess (where each line describes a fixed A). Adopted from [78].

Shifts in this space correspond to six different innovation strategies, based on six value orientations. They are presented, arranged in "the innovation cube", in Figure 6 and explained in detail in terms of some 20 characteristics in [13]. Three of them (1–3) try to squeeze all the possible value out of the existing value box by doing things better. The remaining three (4–6) actually challenge the boundaries: they try to expand the value box, the output is better (doing better things). The distinction between doing existing things better and doing better things is a well-known phrase, which is now being theorized on the basis of three axes, corresponding with the three above: one called technology (in the form of process optimization or product-service innovation), one called resources (utilization optimization or expansion of output), and one called people (better cooperation between people to do old tasks or an expansion of actors who benefit) (Figure 6). The correspondence is as follows. Unity corresponds with technology as a technology must operate as a coherent whole, similarity corresponds with resources as typically resources scale according to parallel and sequenced similarity effects, and pluriformity corresponds with people as it is the dealing with differences in perspectives that determines the required professionalism in people skills, rather than the number of individuals per se. Due to these correspondences, the structure-effort correspondence hypotheses visualized by means of the iso-planes remain intact.

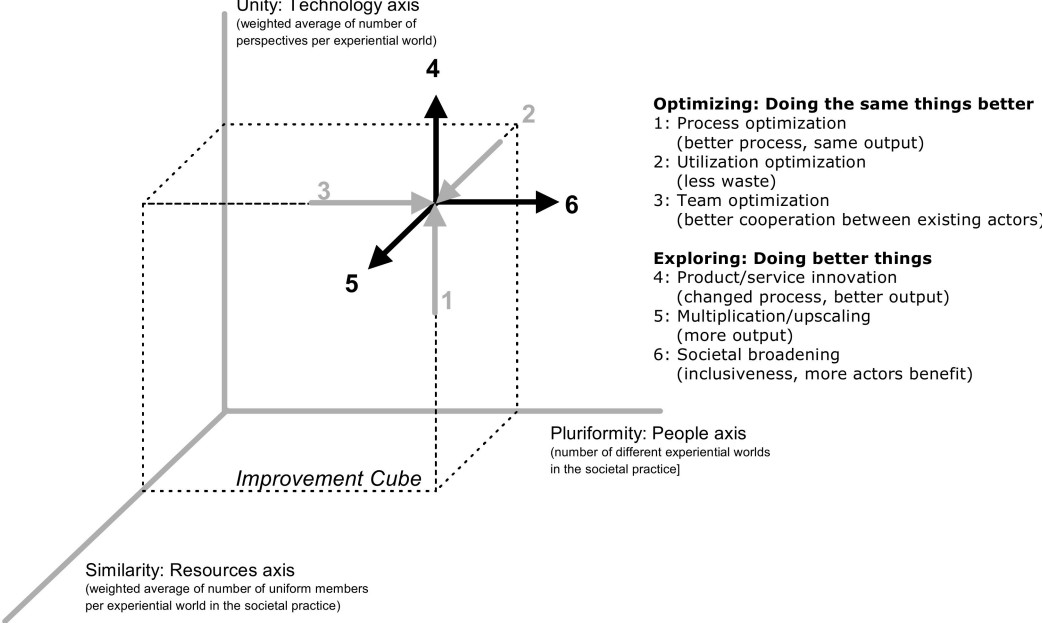

**Figure 6.** The six value orientations of the innovation cube. Adopted from [13].

Societal innovation depends on getting better in all innovation strategies on all axes and therefore requires combining the six value orientations. It should be noted, that system improvement is unable to meet the potential of radical societal innovation, as it only uses the three within-box value orientations (1–3), and neglects the exploring, boundary-challenging triplet of value orientations (4–6). Limiting societal innovation to the three optimizing value orientations (1–3) therefore puts absolute and hard limits on the innovation potential. Typically, the three explorative value orientations (4–6) are called for in times of change, as they seriously question and change existing boundaries. They are relevant predominantly at the start of and in the first half of the life cycles of new societal practices. The three optimizing value orientations (1–3) are called for in the second half and at the end of life cycles when the value box is clear, and the changes become smaller. In terms of transition, therefore, a substantial role should be given to the explorative value orientations, which seek the societal practices of the future. On the long run, changes from predominantly exploring towards predominantly optimizing

modes of innovation will show themselves, resulting in the well-known cumulative bell curves (s curves) used to depict idealized transitions.

*4.5. Societal Innovation as a Systemic Type of Innovation Requiring System Building and Design Thinking*

In this section, we further examine the notion of societal innovation. The term societal innovations was introduced in a report by Diepenmaat and te Riele [79] about societal networks for innovations with sustainability benefits. It is used as the central concept of the inaugural speech of Jan Rotmans, who defined it as a societal renewal process (involving socio-technical change but not limited to it). In superseding functionalist understandings of value, societal innovation is less functionalist than system innovation (as used in the transition literature), but until recently the term lacked a clear definition and theorization in terms of recursive perspectivism. Societal innovation involves social innovation in the form of cross-sector partnerships (resulting in new value chains) and possibly changes in ownership (energy cooperatives for renewable energy to heat and powerhouses). Initially, it runs up too many barriers, having to do with old mindsets and the competition from well-developed but unsustainable products and services. Due to this, it requires active encouragement (pooling of resources and support from government agencies, regulators and knowledge institutes) over an extended period. Its attractiveness lies in serving goals and needs that are presently not well-served (nature preservation or regeneration, and immaterial needs of fairness, care for the well-being of others and the promotion of human flourishing).

The strategy framework for system building of Planko [72] (discussed in Section 3) may help prospective innovators determine collective actions aimed at technology development, market creation, social-cultural change via coordinated action and constituency building, all of which is necessary for societal innovation. Design thinking can help to find "configurations that work" serving the needs of many actors including finance, users and government [65]. According to Ceschin and Gaziulusoy [85], design thinking has expanded from green design to higher systems levels, with active consideration of user needs and life-cycle thinking in eco-design, emotionally durable design (based on personification by the users themselves) and design for sustainable behavior. Broader approaches include design for social innovation and socio-technical system innovation where the latter focusses on "transforming systems by actively encouraging development of long-term visions for completely new systems and linking these visions to activities and strategic decisions of design and innovation teams" where "achieving these visions will require design and innovation teams to use a combination of the approaches in lower levels and use in development of new technologies, products and services (Level 1), new business models (Level 2), new social practices (Level 3) that can be part of the envisioned future systems" [85] (p. 31) (Figure 7).

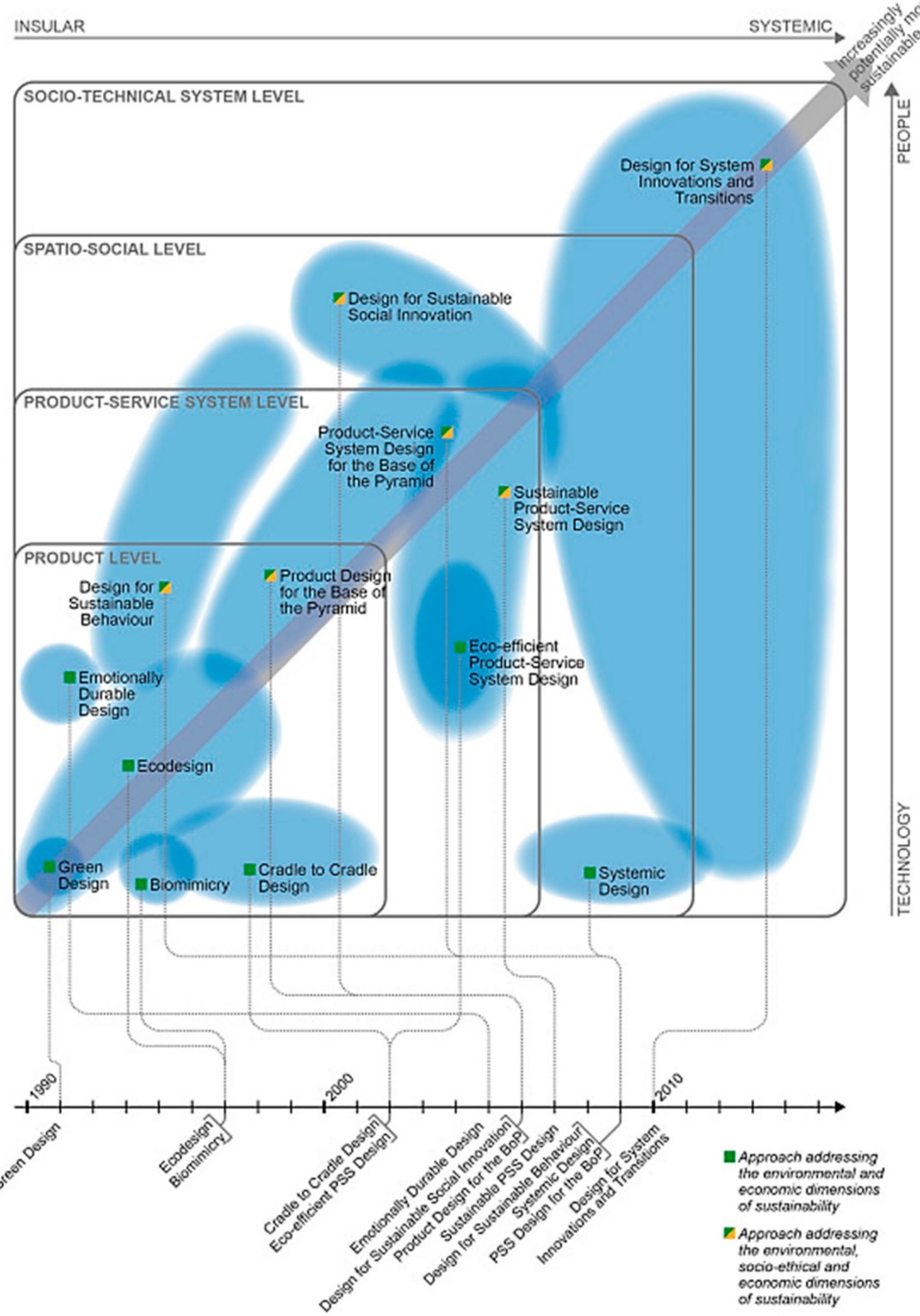

**Figure 7.** The Design for Sustainability Evolutionary (DfSE) Framework with the existing DfS approaches mapped onto it. The timeline shows the year when the first key publication of each DfS approach was published. Adopted from [85].

Design thinking is a prominent element of the RIO approach, a participatory design-based approach for doing reflexive modernization [86], a process of modernization that is mindful of negative side-effects and tries to avoid those [87]. Starting from the idea that system innovations involve changes in action as well as structure, it makes the design of both technical and social features of

societal systems for production and consumption as its central activity and the focus of deliberation. The actors involved include those who are needed for the implementation of the solution.

Bos et al. [88] (p. 140) define the following key aspects of RIO:

○ A focus on concrete design to deal with a specific issue in a specific action context,
○ Systematic reflection on the current structural arrangements of the system at hand and the needs of key actors involved
○ Systematic assessment of needs, values and competencies of the actors involved
○ Connection of identified needs, values and competencies with technical and structural aspects of a socio-technical design and relevant behaviors.
○ Determination of functional requirements and use of morphological diagrams to guide design thinking with attention been given to local circumstances
○ Anticipating structural change and identification of barriers at the regime level that may hinder niche formation
○ Making proposals and actual interventions in order to lower or remove barriers at the regime level
○ Pilots and trials for use with activities of participatory evaluation

The link with recursive perspectivism is that it works with the improvement perspectives of real actors and brings all relevant actors into play with each other. The notion of recursive multi-actor intentionality lifts improvement processes to intentional societal networks at societal scales. This results in a radical and intentional multi-actor interpretation of societal practice, that will enable us to better understand and further (even substantive) societal practices

## 5. Societal Innovation as a Rebalancing of Society

Societal innovation requires collective action from stakeholders in the form of system building activities for societal innovation. Through societal innovation, based on multiple value creation, external costs are being prevented or reduced because of innovation-oriented explorations within a wider frame (a societal improvement perspective), ascertained by the actors. This requires a change in value orientations and fitting multi-actor arrangements with important roles for technological and social innovation, diffusion and institutional change. In this section, we examine the element of social innovation and rebalancing of society that are necessary for societal innovation and an associated remaking of society. Societal innovation seeks multi-actor improvement perspectives that result in better societal practices. A better societal practice is more societally complete: all actors supporting each other in the societal practice and required for the viability of this practice are present, and their interests are met (which is a structural requirement rather than that it requires shared values to a large extent) [79]. Externalization of costs and misery is prevented through design thinking and collaboration. Given the difficulty of dealing with external costs afterwards (because of resistance from producers and consumers negatively affected by corrective measures), it can be deduced that societal innovation is critical to achieving sustainable development and that sustainable development cannot be achieved without societal innovation. A reactive approach of limiting the side-effects of present technologies and practices should be complemented and ultimately replaced by an opportunity-based approach of innovation. This always has been the portée of the sustainability transition discourse but up until now lacked a theorization in terms of (recursive) improvement perspectives and business model innovation.

Continuing too long with just one value orientation that initially generates value, sooner or later inevitably leads to value diminishing, and a forced to switch tracks. This value paradox is a rather depressing rule of thumb. Every positive direction eventually will dry up without any exception. The reason is that crossing the diagonal, where cooperation and coherence are best, ends up in the boundary areas of the perspectivist space. Technological development is an important ingredient for societal innovation (see the technology axis of the innovation cube) but cannot meet the requirements in isolation. The reason is that societal innovation requires complete partnerships of actors willing and

capable to work on societal innovation. Creating such partnerships involves, on top of technological innovation, (a) social innovation, (b) design thinking of how different resources and improvement perspectives can be combined and (c) boundary work to bridge the different experiential worlds and associated ways of thinking amongst relevant stakeholders (actors).

The role of the government differs per case. In the case of Rondeel eggs, the government played a crucial role in funding a design-based transdisciplinary innovation project (which led to Rondeel eggs) and by adapting the local permit system to this innovation which required an unusual amount of space (Figure 8). The government also has an important role to play in the case of sustainable packaging. After relying on industry efforts, it is moving to a more active position, because of a.o. the attention to ocean plastic pollution. The need for change is expressed politically by national governments and the EC. At present steering is being increased, but this may emerge as something needed to make more progress on reducing ocean plastics pollution. A sign of this is that in 2019, "The European parliament has voted to ban single-use plastic cutlery, cotton buds, straws and stirrers as part of a sweeping law against plastic waste that despoils beaches and pollutes oceans", a vote which does not engage with mechanisms for delivery (as a fundamental weakness which underlies political symbolism), but perhaps "paves the way for a ban on single-use plastics to come into force by 2021 in all EU member states" [89]. Helped by the government through special schemes and general innovation support, the sector is involved in the search for plastics recycling and alternatives for single-use plastics. An example support scheme is the UK £20 million Plastics Research and Innovation Fund (PRIF) which aims to explore new ideas and innovations that bring changes in the UK's plastics manufacturing and consumption patterns. Next to this, in many countries, local governments are involved in plastics collecting schemes (which are recycled or burned in waste to energy plants). All these activities are intermediate steps in our long-term societal development from autarkic economies via linear economies, recycling economies and circular economies towards economies in which sustainability is intrinsically valued [13].

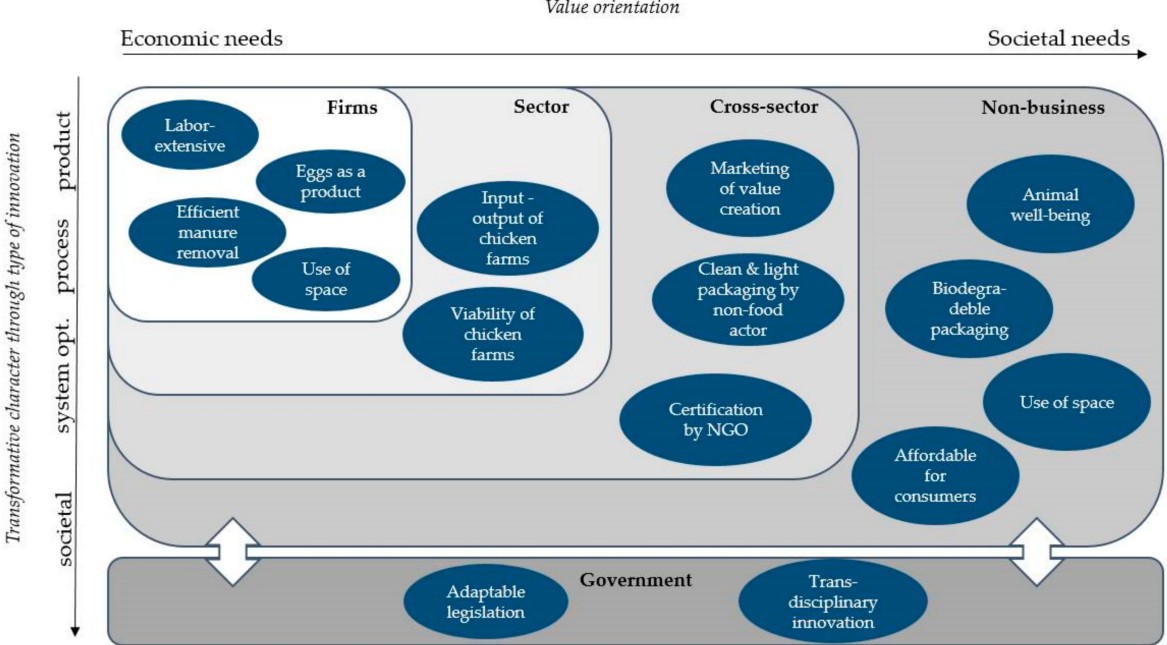

**Figure 8.** Value orientations and transformative character of change for the case of Rondeel eggs (Own illustration).

For working on societal innovation, the Backbone scheme of [13] with governance, societal coordination, arrangements and transactions as interdependent layers offers a useful model for dealing with the challenge of finding suitable transactions through emergent steering [71]. It expands the

focus on transactions to a wider focus (Figure 9), which helps to think about attractive transactions, the development and support for attractive arrangements, and effective governance simultaneously. The Backbone makes the complex and obscure multi-actor context more transparent and manageable [13,90]. It shows how scaling up sustainable transactions requires the presence of attractive arrangements, and that developing these attractive arrangements (operational societal practices) especially in case of sustainability require directional support and guidance, legitimized and financed by actors, such as government and bodies especially established for transition processes.

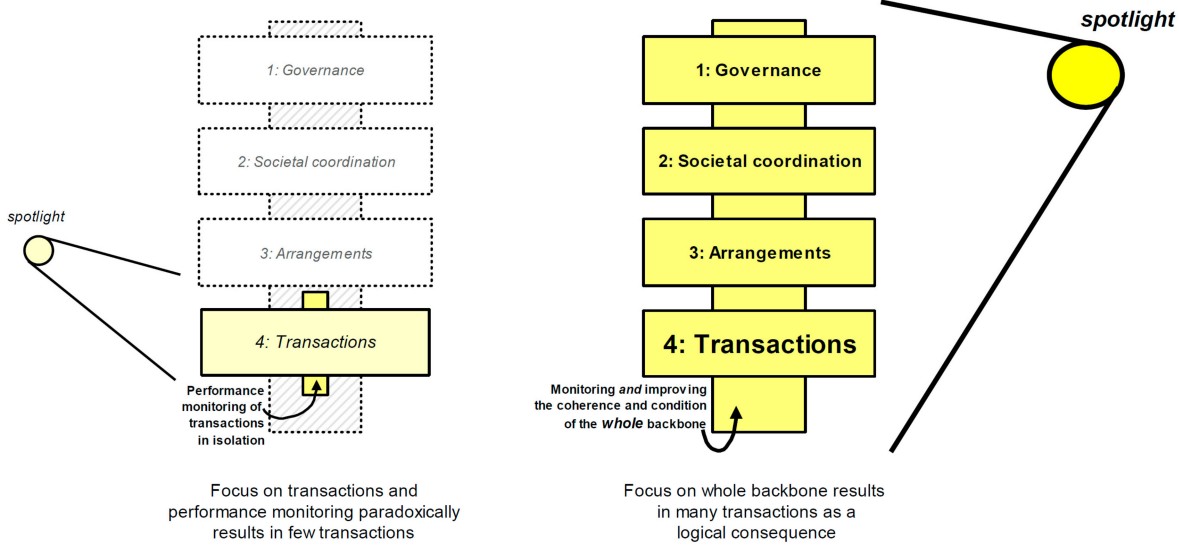

**Figure 9.** The backbone of societal innovation. Adopted from [90], see also [13].

In essence, any transaction has its root cause in benefits for (in needs and wishes of) citizens-consumers, a notion that is not well reflected by the well-known triple helix model, where the focus is on governments, businesses and the knowledge infrastructure, and citizens-consumers and intermediaries are secondary agents. For this reason, developing high-quality backbones requires a penta helix, in which citizens-consumers, business life and governments (the three primary societal participants) are supported by the knowledge infrastructure (including universities) and intermediate parties (the latter one focused on boundary work and knowledge brokerage) in their attempts at societal innovation. Additionally, as a related point, the backbone model for societal innovations helps to bring into focus the important issues of (1) values for justification, and (2) valuation for decision-making [91].

Of course, the government cannot be involved in the promotion of every societal innovation, but it helps to think about the existing policies (whether these are helpful or unhelpful for societal innovations) and new policies and policy adjustment from a different perspective (more innovation-oriented). Active government involvement is needed when existing regulations create unnecessary barriers to societal innovation, when the level playing field is heavily skewed towards the status quo and when collective action problems block market formation. Most sustainability innovations fall within these categories, especially those that are disruptive (transformative). Societal innovations are important vehicles for socio-technical transitions and socio-economic transformations, and therefore for sustainable development. Up until now, this is poorly understood and accepted. The attention to alternative forms of well-being in welfare measurement is a positive step for achieving societal innovations (such as nature-inclusive agriculture or a social circular economy), but in itself insufficient. Multiple value creation should become a topic for science, for stakeholders (including government and political parties) and a topic for discussions on modernity [92]. When this happens, the process of modernity may become more pluriform in combining the experiential worlds and improvement perspectives of different actors, resulting in plurimodernity [13]. Since negative value is anticipated and proactively dealt with, plurimodernity involves less unintended negative effects. As a result, modernity will

take a less mono-sectoral course. For reasons discussed in Section 4, multiple value creation across sectors requires social innovation (in terms of who is involved and how) and settlement mechanisms for dealing with multiple benefits and costs with careful attention being given to place-specificity, the improvement perspectives of those involved, institutional logics and arrangements in need of change or creation and root causes of problems. At present social innovation, as a key component of societal innovation, is viewed very much in functional terms, as an aid of technical change, but it needs to be viewed in another sense too: as a vehicle to remake or rebalance society [16,93]. Transformative social innovations are oriented at precisely that [77].

The barriers to societal innovation are formidable. As an example, the Hercules project aimed at structural changes in both animal and crop production, relapsed from an effort for reflexive modernization to ecological modernization, by ultimately leaving the structural features of the socio-technical regime intact [86]. The difficulties encountered are said to be typical for projects aiming at reflexive modernization. This underscores the conclusion of Kivimaa and Kern [94] that policy mixes for transitions should include elements of 'creative destruction', involving both policies aiming for the 'creation' of new and for 'destabilizing' the old. Societal innovation depends on costing negative externalities, something for which the government has a crucial task.

Sustainable development requires societal innovation, which aims at the development of societal improvement perspectives. The most intriguing feature of well-coupled societal improvement perspectives is that they have self-executing qualities. We only have to develop them, and they will unfold themselves, driven by intentional logic. Further progress will depend on helpful new circumstances, learning economies and coordinated actions. Aiding and professionalizing societal innovation, therefore, is an important prerequisite for sustainable development. It is important to take seriously the value frames and intentional logics of the actors involved. However different they are from one's own. In this respect, prescribing SDGs to a society that is oriented elsewhere (for example to profit-seeking and consumerism) is like flogging a dead horse (leading to ticking the box exercises and an SDG-based relabeling of activities).

## 6. Conclusions

In this paper, we examined innovation from different vantage points: a business model vantage point, a boundary-spanning vantage point, a sustainability transition vantage point and the vantage point of recursive perspectivism. Recursive perspectivism is used to identify six types of innovation, based on six distinctive forms of value creation, and in combination able to support societal innovation. In the paper, we argue that societal innovation (based on multiple value creation) is needed for sustainable development, which moves innovation activities away from a weak or pale greening of business activities (with the help of CSR) towards innovation activities in which environmental protection and societal value is internalized instead of externalized. This takes deliberate efforts and involves many difficulties but is necessary because the current approaches for working towards sustainable development (via CSR1.0 and weak environmental policies) fall short of what is needed [19,21,23].

Societal innovation involves exploration, cross-sector collaboration, changes in boundary conditions, the emergence of new business models (based on multiple value creation) and the recreation of modernity (each of which is necessary for the other aspects to happen and to continue). It is not directed at minimizing the damage resulting from existing societal practices (optimizing the existing), but at the development of adapted or even new radically better societal practices preventing damage (creating the new). When properly done, societal innovation addresses root causes of unsustainability (social and institutional conditions that allow for the externalization of costs to society, the unprofitability of (disruptive) sustainability business practices, and regime actors opting for improvement of existing systems and practices rather than the creation of new ones).

Rather than calling for the identification of shared values as the basis for sustainability action (as in the framework of Porter and Kramer [40]), our methodological framework for societal innovation makes a point of understanding and respecting differences and bringing these into coherence as flexible

components in intentional collective multi-actor networks. The only thing that needs to be shared at the level of societal practices is the acknowledgement that actors require *each other* in realizing their *own* needs and wishes and may help each other in this respect. Contextual aspects enter via the improvement perspectives. Different contexts will give rise to different improvement perspectives (based on what an actor thinks it can do and has reasons to value). Alignment of improvement perspectives will depend on intermediation and trust and thus on external factors (which differ across contexts) and whose influence requires further research (to arrive at conclusions about the role of context).

Societal innovations guide and prepare socio-technological transitions and socio-economic transformations towards more societally complete practices. For this reason, they deserve more attention than currently is given to them, by business scholars and by sustainability transition scholars. With this paper, we hope to advance the research agenda on societal innovation based on multi-actor improvement perspectives and associated intentional logics, as topics that are weakly theorized in the business literature on sustainable development and the sustainability transition literature.

**Author Contributions:** Overall, the paper rests on the merging of many different lines of thought of the three authors, familiar with the sustainability debate and sustainability research. The the awareness that more societal notions of innovation are required for sustainable development, and the discussions and conclusions with respect to the need for societal innovation were a joint insight of the three authors. Sections 1, 5 and 6 were jointly written. M.V. authored Section 2, R.K. wrote Section 3 and H.D. is the lead author of Section 4 about the the five layered methodological framework, recursive perspectivism, the concepts of improvement perspectives and intentional logics, the theories and methods for societal innovation built on this (specifically the innovation cube and the six innovation types, the backbone, PAIR analysis and intentional multi-actor modelling). All authors have read and agreed to the published version of the manuscript.

**Funding:** The APC was funded by MSI.

**Conflicts of Interest:** The authors declare no conflict of interest.

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
