# Peer review of "Why Sustainable Development Requires Societal Innovation and Cannot Be Achieved without This"

_sustainability, doi:10.3390/su12031270_

Round 1

Reviewer 1 Report

The paper is interesting, presenting new viewpoint on a links between business innovations, societal innovations and sustainability. It lays a good foundation for further discussions, based on above mentioned concepts and in redefining economic actors goals in general. 

Although, these ideas need some refinement in order to be universally accepted as a new theory. First, it needs some adjustment in order to fit the different societies, there cultural background and overall values are more individualistic (eg. USA) and their artificiall congruence till the form of widelly accepted common higher goals (regulative norms) can induce new challenges. 

These ideas also should be very carefully presented in most of post-Soviet societies, as there well known extreme social experiments and elevation of social norms and (as it appeared in that day) some form of societal innovations above all other (individual, business and etc.) interests took place causing very negative effects.

So, in my opinion, the additional section describing the suitability of presented ideas in different societies would add more scientific soundness to the manuscript and would allow it to more substantially contribute to the development of the science.

Author Response

Dear Reviewer,

Please kindly find the attached file in response to your comments.

Kind regards

Reviewer 2 Report

Comments to the Authors

I have read the assigned manuscript with interest. The paper is focused on sustainable development within the societal innovation framework. The major aim is to contribute to the literature adding different perspectives to explain this “new” phenomenon.

The paper represents a nice attempt to explore sustainable development in relation with societal aspects, therefore I think is in line with the aim of the journal. However, it needs some minor adjustments and proofreading to be considered for publication. 

Please find enclosed my comments.

Some of the sentences are quite long and not easy to read in the current status (for example the last part of the abstract). Moreover, there are some acronyms that have to be explained (TIS at page 9 and RIO at page 22).

At page 10, line 373, I would rephrase the sentence “The section is based ... co-author”. The reference on “co-author” is without year, so it is difficult to understand how Dr. Diepenmaat is contributing to this literature.

Finally, at page 17 there is an erased sentence to be removed from the text.

There are some references missing in the text: page 2 line 64, I would have added some references when talking about the triple helix model; same page, line 75, when “those literature” is mentioned, could you please add some references about it?; page 4, line 132, when citing UNEP (2001) there is no page reported, please add; Table 1 page 14, are these examples from authors’ elaborations as it seems from the text? If this is the case please make it clear in the footnote; finally, the bullet points at page 23 when referring to Bos et al (2009), please move “(p.140)” close to the reference so it should become Bos et al. (p. 140, 2009).

Finally, some clarification throughout the text. Could you spend a couple of words when you refer to modernity and plurimodernity literature at the end of page 2?; at the end of paragraph 3, after explaining the various aspects of  sustainability and value creation, you might add some sentences to conclude this argument; same for page 11 after the last three lines 400-402.

Good luck!

Author Response

(The authors gave the same response as above.)

Reviewer 3 Report

I am grateful for the opportunity to become familiar with this manuscript which has great chances to become an interesting article contributing to the field of Sustainability, innovation and governance. In order to make the paper even better, here are my few  remarks:

I would suggest to have a more scholarly title. At present the title sounds slightly as a (self-)marketing or a think-tank paper title, hence, not scientific enough in my opinion. The abstract of the article does not seem to correspond to the IMRAD scheme. An abstract should briefly, concisely and clearly summarize basic goals, methods, results and conclusions of an investigation. The structure should follow more exactly the IMRAD scheme and highlight major findings. When searching a database, the abstract is a basis for our decision to download the paper, read it and – maybe - quote it. The authors list a number of actors that they address, but somehow neither SMEs (small-medium enterprises) nor TNEs (transnational enterprises or transnational corporations) are part of the picture. See eg the lines 45-55:  ' [...] the concerns of citizens, workers, local government and special interest groups [...] have to be considered'. Is there a particular reason why the authors omit completely the private sector actors--both TNEs and SMEs--here, and if so why? Maybe fleshing just a phrase would help clarifying this issue. The text is better be edited as I noticed a few typos, misspelling, etc.

Author Response

(The authors gave the same response as above.)

Round 2

Reviewer 1 Report

Although a paper was improved, I still have doubts about some of the fundamental statements, on which this research is based. You argue, that: "The only requirement of an intention is that it
represents an improvement in an environment (in any desired sense) according to the actor at stake. A growing intentional logic will cause the actor to rise and act." It may be true, may be not. All human beings are different, some are diligent and the surrounding environment and social norms are motivating humans to work, create, take care of traditions and environment (eg. Japan) in some societies with different cultural, historic and economic background it is hardly acceptable and proves negative results (early mentioned Soviet Union, in some form Haiti and etc.). 

In my opinion, You should specify more the context and lay less confidence on presumptions that can (or may) be argued.

Author Response

Dear Reviewer,

Thanks for your further comments and please kindly find the attached pdf for our response.

Round 3

Reviewer 1 Report

Authors reacted to all my comments and improved the paper accordingly. I suggest publishing it in a current form.